# The Immunobiology of Nipah Virus

**DOI:** 10.3390/microorganisms10061162

**Published:** 2022-06-06

**Authors:** Yvonne Jing Mei Liew, Puteri Ainaa S. Ibrahim, Hui Ming Ong, Chee Ning Chong, Chong Tin Tan, Jie Ping Schee, Raúl Gómez Román, Neil George Cherian, Won Fen Wong, Li-Yen Chang

**Affiliations:** 1Department of Medical Microbiology, Faculty of Medicine, Universiti Malaya, Kuala Lumpur 50603, Malaysia; yvonneljm@um.edu.my (Y.J.M.L.); ainaaibrahim@um.edu.my (P.A.S.I.); onghuiming@um.edu.my (H.M.O.); chongcheening@um.edu.my (C.N.C.); wonfen@um.edu.my (W.F.W.); 2Deputy Vice Chancellor’s Office (Research & Innovation), Universiti Malaya, Kuala Lumpur 50603, Malaysia; 3Division of Neurology, Department of Medicine, Faculty of Medicine, Universiti Malaya, Kuala Lumpur 50603, Malaysia; chongtin.tan@gmail.com (C.T.T.); scheejieping@hotmail.com (J.P.S.); 4Vaccine Research and Development, Coalition for Epidemic Preparedness Innovation (CEPI), Askekroken 11, 0277 Oslo, Norway; raul.gomezroman@cepi.net (R.G.R.); neil.cherian@cepi.net (N.G.C.)

**Keywords:** henipavirus infections, encephalitis, chiroptera, innate immunity, humoral immunity, cellular immunity, interferon type I, animal models, medical countermeasures

## Abstract

Nipah virus (NiV) is a highly lethal zoonotic paramyxovirus that emerged in Malaysia in 1998. It is a human pathogen capable of causing severe respiratory infection and encephalitis. The natural reservoir of NiV, Pteropus fruit bats, remains a continuous virus source for future outbreaks, although infection in the bats is largely asymptomatic. NiV provokes serious disease in various mammalian species. In the recent human NiV outbreaks in Bangladesh and India, both bats-to-human and human-to-human transmissions have been observed. NiV has been demonstrated to interfere with the innate immune response via interferon type I signaling, promoting viral dissemination and preventing antiviral response. Studies of humoral immunity in infected NiV patients and animal models have shown that NiV-specific antibodies were produced upon infection and were protective. Studies on cellular immunity response to NiV infection in human and animal models also found that the adaptive immune response, specifically CD4+ and CD8+ T cells, was stimulated upon NiV infection. The experimental vaccines and therapeutic strategies developed have provided insights into the immunological requirements for the development of successful medical countermeasures against NiV. This review summarizes the current understanding of NiV pathogenesis and innate and adaptive immune responses induced upon infection.

## 1. Introduction

In the events of the recent COVID-19 pandemic, more attention and effort has been devoted to studies and research on potential pandemic-causing pathogens, one of which is the Nipah virus (NiV). NiV is an emerging paramyxovirus with a high pathogenicity that has been causing near-annual outbreaks in the South Asia region since its discovery in Malaysia in 1998 [1]. It is currently listed as one of the top 10 emerging viruses that require urgent research and development in public health emergency contexts by the World Health Organization (WHO) [2], and it has been made a priority for vaccine development by the Coalition for Epidemic Preparedness Innovations (CEPI) [3] and the United Kingdom Vaccine Network [4].

NiV is a negative-sense single-stranded RNA enveloped virus and is a member of the genus Henipavirus in the family Paramyxoviridae [5,6]. The genome of the virus is non-segmented and is approximately 18 kb nucleotides long [6,7,8,9]. The viral genome encodes six structural proteins: nucleoprotein (N), phosphoprotein (P), matrix protein (M), fusion glycoprotein (F), attachment glycoprotein (G) and the RNA polymerase or large protein (L). In addition, there are three accessory proteins within the P: the V, W and C proteins, as a result of mRNA editing and the alternative start codon. Overall, NiV genome sequence analyses have identified two main clades: the M genotype, which comprises the Malaysian NiV isolates (NiV-M), and the B genotype, which includes Bangladesh (NiV-B) and India NiV isolates (NiV-I) [7,10,11]. Despite the three strains sharing a high percentage of homology (NiV-M and NiV-B strains share 91.8% homology, and NiV-I sharing 85.14–96.15% homology with both NiV-M and NiV-B), the B clade infections were shown to be significantly more pathogenic than the M clade [10,12,13,14].

In the Malaysia outbreak, NiV infection was characterized as a respiratory and neurological disease that resulted in over 250 cases and fatalities exceeding 100 cases [1]. Besides Malaysia, NiV was reported in neighboring countries such as Singapore, the Philippines and South Asia (Bangladesh and India). In South Asia, cases were reported almost annually, with the most recent case of NiV infection reported in 2021 in Kerala, India [15]. Clinically, respiratory infections were more common and the mortality was higher among the NiV cases reported in Bangladesh and India, as compared to the cases in South East Asia. The variation in severity of symptoms and mortality between NiV cases in both regions could be attributed to the different genetic makeup of the strains pervasive in either region, or the differences in access and quality of medical care between the two regions [16]. Nevertheless, NiV infections are generally associated with acute respiratory distress, encephalitis and in some cases myocarditis [17]. Additionally, some patients experienced drowsiness, extreme lethargy, mental confusion and in the worst cases coma. In fact, a percentage of patients experienced residual neurological complications, such as late-onset encephalitis, years after the initial infection [18,19].

Fruit bats of the Pteropus genus have been suggested as the natural reservoir for NiV. The bats harboring NiV remain asymptomatic [20,21,22] and therefore could facilitate the spread of the virus to susceptible hosts during spillover events. In humans, epidemiological studies have implicated animal-to-human and human-to-human transmissions as the main routes of NiV spread; the former was connected to exposure to infected animal fluids such as its saliva, urine and excreta, whereas the latter was connected through contact with body fluids from infected individuals, specifically via respiratory droplets [20,23,24]. The route of NiV transmission in the Malaysia and Singapore outbreaks was identified to be animal-to-human, whereby bats harboring NiV transmitted the virus to pigs through direct contact, which then acted as amplifying hosts and subsequently transmitted the virus to humans via aerosol droplets [1,5]. Meanwhile, in the Bangladesh and India NiV outbreaks that occur almost annually, transmission of the virus is also animal-to-human, but via ingestion of food or fluid contaminated by NiV-infected bats or via direct contact with NiV-infected bats. Besides this, the human-to-human transmission of NiV was also reported, and this was a common mode of transmission in Bangladesh, comprising half of the NiV cases reported between 2001 and 2007 [25,26,27]. Sociocultural expectations to care for ill family members, poor infection control practices and lack of healthcare resources are factors that could have contributed to the higher number of human-to-human NiV transmissions in Bangladesh relative to Malaysia [24,26].

Despite NiV outbreaks occurring almost annually and the pandemic potentiality of the virus, no vaccines or therapeutics have yet been approved and made available for human use [28,29,30]. Vaccines in general are targeted to induce humoral immunity, specifically protective antibodies; recent vaccine development also aims to generate cellular immunity. This is because both immune subsystems are crucial to provide an effective immune response towards the infection and for protection against the disease. However, inadequate clinical specimens available for in-depth analysis due to no following NiV outbreak in Malaysia and small sporadic outbreaks of NiV in Bangladesh and India result in constraints to the recapitulation of clinical signs of the human NiV disease, as well as the monitoring and evaluation of the immune response following NiV exposure. Hence, this review aims to investigate and integrate the findings of both the innate and adaptive immune responses towards NiV infection to better understand how the immune system in humans and across animal species could lead to a mechanism for viral escape.

## 2. Methods

A targeted literature search was conducted using the digital archives Pubmed, Google Scholar and ScienceDirect with “Nipah” OR “Hendra” OR “henipaviruses” as keywords with additional MeSH terms: “Nipah virus infection”, “Innate immune Nipah virus”, “Adaptive immune Nipah virus”, “B cells Nipah virus”, “T cells Nipah virus”, “Epidemiology Nipah virus”, “Clinical features Nipah virus”, “Diagnosis Nipah virus”, “Surveillance Nipah virus”, “Vaccine Nipah virus”, “Monoclonal antibodies Nipah virus” and “Animal model experiment Nipah virus”. All literature reviews, original papers and case reports referring to aspects of NiV origin, mode of transmission, clinical presentation, pathogenesis and immune responses published, until 31 March 2022 were included. The cross-references from these publications were also included. Additionally, epidemiological reports from the WHO, CEPI and other public health organizations were assessed. The search strategy, shown in Figure 1, was performed with the aim of finding literature describing the immune responses, pathogenesis, transmission of the disease in animal models and medical countermeasures associated with NiV.

## 3. Replication Cycle of NiV

The NiV particle has six structural proteins, namely the N, P, M, F, G and L, which are arranged accordingly in the RNA genome from 3′ to 5′ (Figure 2) [6]. The replication cycle of NiV starts when the virion attaches to the host cell receptors, ephrin-B2 and -B3 via the NiV G protein [31,32,33]. Next, the NiV F protein mediates the fusion of the viral envelope with the host cell membrane, releasing the viral genome into the cytoplasm. The viral genomic RNA is associated with N, P and L proteins, which forms the ribonucleoprotein complex and is involved in the transcription and replication of the virus. The L polymerase catalyzes the transcription of the virus genomic RNA into mRNAs for protein translation. The translated viral surface glycoproteins F and G are inserted into the host cell endoplasmic reticulum for post-translational modifications, particularly glycosylation. The other translated viral proteins—N, P, M and L—remain in the cytoplasm. When abundant viral mRNA transcripts are produced, full-length anti-genomes are then synthesized to generate more copies of the NiV genome. These new copies of genome assemble with the viral proteins near the host cell membrane where F and G proteins are studded, and the budding of new virions facilitated by the M protein will occur.

## 4. Pathogenesis of NV

NiV enters through the oronasal route into human and other animal hosts to cause an infection. The virus infects the epithelium cells along the respiratory tract, and a high concentration of viral antigens could be detected in the lymphoid and respiratory tissues [12]. Initial viremia then spreads the virus to other parts of the body, while secondary replication occurs in the endothelium. The NiV infection of host cells starts when the viral G protein attaches to the cellular receptors ephrin-B2 and -B3 [31,32,33]. The virus then rapidly disseminates to different organs, including the spleen, kidneys, heart and liver within the first week of infection [14,34,35]. Both ephrin-B2 and -B3 are found on a wide range of cell types including epithelial and endothelial cells, as well as neurons. Both these cellular receptors are highly conserved across animal species, which explains the broad species and tissue tropism of NiV [36]. Interestingly, a recent study has observed that smooth muscle cells that lack the cellular receptors ephrin-B2 and -B3 were permissive to NiV infection and produced high viral titers similar to permissive cells expressing the cellular receptors [37]. There was prolonged NiV production in the smooth muscle cells with no cytopathogenic effects. Together, the study suggested the likely existence of an unidentified entry receptor for NiV or a non-specific virus entry mechanism. Besides, NiV was also reported to enter and infect the central nervous system via circulating immune cells, specifically immature dendritic cells and monocytic cells [38]. These cells were noted to be NiV-permissive; however, the virus did not replicate efficiently in them. Nevertheless, the NiV-infected immune cells migrated across the in vitro blood–brain barrier and infected susceptible cells in a focused manner, similar to observed neuronal infection and the presence of focal lesions in the brain of both NiV-infected human and animals [39,40].

## 5. NiV F and G Glycoproteins

There are two distinct NiV surface glycoproteins that play essential roles in the entry of NiV into host cells: the G protein, which is responsible for host cell receptor-binding, and the F protein, which mediates membrane fusion between the virus and the host cell [6]. Unlike other paramyxoviruses, the NiV G protein is unique due to its inability to function as hemagglutinin and neuraminidase [32]. Instead, the protein binds to host cell receptors ephrin-B2 and -B3 for virus entry. As a type II membrane protein, the G protein exhibits characteristic tetramerization through its N-terminal α-helical stalk domains, while its globular head domain at the C-terminal binds to the host cell receptors [41,42]. On the other hand, the NiV F protein occurs in trimeric form and belongs to the class I viral fusion protein. It possesses a globular head that consists of three domains, and it attaches to the host cell membrane via the C-terminal α-helical stalk [43].

The experimental models of NiV F and G proteins were shown to undergo a series of conformational changes during receptor engagement in order to enhance host cell membrane fusion for virus entry. It was proposed that the binding of the G protein to the host cell receptor triggers conformational changes that separate its head and stalk domains, thus allowing interaction with the F protein, which subsequently triggers its refolding [44]. In the process of conformational changes, one of the four head domains of the G homotetramer protein rearranges its receptor binding site to ephrin-B2, while the other three head domains angle towards the viral membrane [45]. The conformational changes in the protein were suggested to promote receptor engagement. The overall architecture of the G protein assumes a distinctive structural conformation that is different from other paramyxovirus attachment glycoproteins.

In addition to G protein, the NiV F protein plays a crucial role in mediating the fusion of the viral envelope with the host cell membrane. The F protein undergoes conformational changes both pre- and post-fusion in order to insert its hydrophobic fusion peptides into the host cell membrane, which is commonly observed in class I viral fusion proteins [46]. Most evidence shows that a direct and specific interaction occurs between the viral F and G proteins prior to host cell receptor binding, which is necessary to activate membrane fusion [47]. However, a recent study did not detect the interaction of NiV F and G protein ectodomains, and proposed that the F and G proteins did not form a stable complex on the cell surface before ephrin-B2 activation [48]. The interaction between the F and G proteins could likely be dynamic and transient; hence, further investigation on the mechanism of NiV fusion will help to resolve the viral fusion system.

### NiV F and G Glycoproteins as Therapeutic Targets

The approaches currently used for the development of NiV medical countermeasures are mainly focused on the NiV F and G proteins. Both these proteins have been identified as the targets of neutralizing antibody responses and have been shown to provide protection in animals challenged with NiV [49,50]. More potent neutralizing antibodies were induced by pre-fusion-stabilized F protein as compared to post-fusion F, implying that the stabilization of the pre-fusion conformation of the protein is necessary to increase immunogenicity [51]. As for the G protein, the head domain was discovered as the target for neutralizing antibodies in rhesus macaques vaccinated with tetrameric NiV G ectodomains [45]. Furthermore, multimeric forms of the G protein were found to elicit higher neutralizing antibody titer in mice as compared to the monomeric protein [51]. In addition, cross-reactive monoclonal antibodies from a human donor, who had prior history of inoculation with Hendra virus (HeV) vaccine, were found to afford post-exposure protection against both NiV-M and NiV-B in ferrets [52]. It is postulated that the elicited antibodies target the NiV receptor binding protein head domains, thus competing with the binding of NiV G with ephrin-B2 and -B3 for receptor engagement and inhibiting viral entry. Taken together, from an immune response perspective, the NiV F and G proteins are particularly important as target antigens to trigger the production of NiV-neutralizing antibodies for the development of NiV medical countermeasures.

## 6. Innate Immunity

The innate immune system is our first line of defense against foreign materials from entering the body (Figure 3). Neutrophils are the first immune cells to be recruited to the site of infection and are armed with several defense mechanisms, including production of reactive oxygen species, antimicrobial peptides and neutrophil extracellular traps (NETs) [53,54,55,56]. NETs are web-like traps composed of nuclear or mitochondrial DNA, antimicrobial peptides and proteolytic enzymes capable of killing entrapped microorganisms [57,58]. Numerous viruses, including severe acute respiratory syndrome coronavirus 2 (SARS-CoV-2) [59,60,61], influenza A virus (IAV) [62,63] and respiratory syncytial virus (RSV) [64,65], have been shown to induce the formation of NETs. However, the biological significance of NETs in the host antiviral mechanism is yet to be fully characterized. Nevertheless, it has been shown that NETs can trap and immobilize viral particles via electrostatic interaction, allowing antiviral molecules located in NETs, such as myeloperoxidase, cathelicidin and α-defensins, to act on the viruses [66]. For instance, myeloperoxidase has a strong antiviral property against human immunodeficiency virus type 1 (HIV-1) [67,68], while α-defensins display virucidal activity on both enveloped and non-enveloped viruses [69]. In respiratory viral infections, neutrophils are seen as a protector against infection, whereas NETs seem to be detrimental to the host. The depletion of neutrophils increases the mortality rate of IAV-infected mice [70], while excessive neutrophil activation and NETs formation cause lung inflammation, which could benefit influenza virus infection [62,71,72,73]. During RSV infection, NETs are able to entrap virus particles and limit virus spreading, but at the same time, they cause airway obstruction in children [74]. Hence, neutrophils and NETs are double-edged swords whose activation and activity need to be tightly regulated to provide a more protective role and less tissue damage to the host.

The battle between the host cell’s ability to activate the innate immune response following an assault and the microbes’ ability to evade and cripple this reaction often determines the outcome of an infection. Henipaviruses, specifically NiV, encode several viral factors that serve this purpose during their viral life cycle. In vitro infection of endothelial cell lines with NiV was shown to induce the production and secretion of several host antiviral proteins, type I interferon (IFN-I), as well as inflammatory chemokines and cytokines [75]. Upon the initial attachment and fusion of NiV to the host cell membrane, cellular cytoplasmic RNA helicases would recognize the released viral genomic RNA and trigger a robust activation of the IFN-I response, which is part of the host innate antiviral defense, along with several IFN-induced antiviral genes such as IP-10, ISG56 and OAS1 [76,77]. This robust upregulation of the IFN-I response upon exposure to NiV is absent in pteropid bats. The pteropid bats, which are an important reservoir for many viruses, have a constitutively active IFN-I system, resulting in the induction of a specific subset of IFN-stimulated genes. It is believed that maintaining a constitutive level of IFN response in the absence of a viral infection bestows bats the ability to control viral replication and coexist with a variety of viruses while remaining asymptomatic to the infections [78]. Additionally, pteropid bats could rapidly induce their type III IFN responses, which are antiviral cytokines, and at the same time maintain a constitutive level of IFN-I response after virus infection This unique regulation of bats’ innate antiviral system could be the key to coexisting with viruses where the immune reaction is activated upon infection, only to sufficiently restrict viral replication and not to achieve viral clearance [79,80].

To counteract this, NiV expresses several structural and non-structural proteins that can distinctively modulate the activation of IFN-I signaling and production at multiple stages within the signaling pathways [81,82]. One example is the NiV V protein which interacts with the cellular signal transducer and activator of transcription (STAT) 1 and 2 proteins, thereby sequestering them from the downstream effector of IFN response: ISGF3 transcription factor complex [83], a prominent modulatory component of the host late phase antiviral response [84]. In addition to STAT 1 and 2 interactions, the NiV V, P and W proteins have also been reported to interact with host STAT 4 and STAT 5 proteins and effectively alter the activity of STAT proteins along with the innate antiviral response [81]. The V protein also interacts with the melanoma differentiation-associated protein 5 (MDA5), an antiviral activator that accounts for the generation of IFN [85]. Upon interaction with MDA5, NiV V protein could then suppress IFN activation by dephosphorylating the MDA5 [86,87]. The antiviral ability of these NiV proteins was further demonstrated using individual NiV P, V and W proteins constructed into recombinant Newcastle disease viruses (NDV) and used for the infection of primary human monocyte-derived dendritic cells [88]. In comparison to the parental NDV infection, which triggered robust IFN-α/β production, the expression of the three individual NiV proteins in NDV successfully subverted the IFN-α/β responses along with reduced cytokine productions. On the other hand, the NiV M protein was reported to antagonize the host IFN-I reaction by promoting the degradation of the Tripartite Motif 6 (TRIM6) protein, which then blocked the synthesis of IFN-I and its signaling pathways [89,90]. The available data strongly indicate that NiV employs mechanisms to promote immune evasion via a complex multifaceted interference approach.

In agreement with in vitro data, the stimulation of host innate immunity and IFN-I signaling was confirmed in vivo using several animal models. In the Syrian Golden hamster model of NiV infection, the production of cytokines, chemokines and IFN-I signaling was detected first in the lung followed by brain tissues, which coincided with the development and progression of the disease [91]. The findings were recapitulated using the ferret model [77]. Moreover, the importance of the innate immune response, particularly the IFN-I response, in the control of NiV infection was demonstrated in mice deficient in IFN-I receptor (IFNAR-KO), which completely lost the resistance to NiV infection that was naturally possessed by the wild-type mice [92]. In congruence with this, the administration of poly(I)-poly(C12U), more commonly known as Rintatolimid (Tradename in US: Ampligen), which induces IFN-α/β production in an NiV-infected animal model prevented death in five of the six infected hamsters [93]. However, the detailed molecular mechanism of poly(I)-poly(C12U) in antagonizing viral replication is currently unclear as the resulting IFN-I and cytokine/chemokine production induced by poly(I)-poly(C12U) could alter the adaptive immune response against NiV and subsequently change the outcome of the infection.

In addition to the antiviral IFN-I response, NiV infection also led to the production of several pro-inflammatory cytokines such as tumor necrosis factor alpha (TNF-α) and interleukin-1β (IL-1β) [91]. Disproportionate production of such inflammatory cytokines could be disruptive to the host as excessive inflammation can contribute to the pathogenesis of virus infections, as noted in infection by SARS-CoV-2 [94]. In the case of NiV infection, the production of TNF-α and IL-1β coincided with the first sign of NiV infection in the brain [91]. The pro-inflammatory effect of TNF-α and IL-1β disrupted the integrity of the blood–brain barrier and contributed to the neurological defects observed in NiV-infected patients [95,96].

Collectively, the balance between host antiviral response and the virus’s ability to neutralize it is critical to ensure successful NiV replication without causing premature host mortality. Similarly, inflammation induced by the infiltration of neutrophils is beneficial for controlling virus replication, but hyper-inflammation from excessive neutrophil activation could lead to undesirable tissue damages. Hence, equilibrium between beneficial and noxious effects of neutrophil-induced inflammation must be preserved for the host to prevail against virus infection.

## 7. Adaptive Immunity

### 7.1. Humoral Immunity—B Lymphocytes

Humoral immunity refers to the antibodies induced in the body upon exposure to antigens. In viral infection, the primary immune response occurs during the first encounter with antigens, and this response can take up to two weeks to develop antibodies specific against the virus [97]. However, if re-infection occurs, the anamnestic response develops rapidly in a day or two, mediated by antigen-specific memory cells. The existence of antibodies specific against a virus in serum of individuals only stipulates the indirect and brief assessment of the humoral immunity and may not precisely reflect the presence of long-term humoral immunological memory, specifically the memory B cells present in the host. Antibody molecules are short-lived with an average life span of weeks, whereas memory B cells are long-lived cells with a life span ranging from several decades to life-long antibody persistence [98]. The latter has been reported following immunizations against smallpox, even after a long period of viral clearance [99]. On the other hand, the antibody titer elicited following measles, tetanus and diphtheria vaccination was reported to decline gradually, where regular booster immunizations are required to sustain protective levels [100,101]. In a recent study on humoral immune response to SARS-CoV-2, neutralizing antibody titer in the serum was noted to reduce over time after the initial infection [102]. Therefore, in addition to monitoring the presence and the level of serum antibodies post-infection, the determination of the population and diversity of B cell repertoire would be beneficial to provide a comprehensive understanding, particularly the long-term humoral immunity and protective capacity of antibodies specific against the virus, including during the event of re-exposure.

In general, during a humoral immune response triggered by antigens, antibody secreting cells (ASC), also known as plasma B cells, will be generated through the differentiation and proliferation of naive B cells (Figure 3). IgM antibodies are the first-generation antibodies produced by ASC, followed by the occurrence of class switching to immunoglobulin (Ig), generating ASC and memory B cells that are able to produce other classes and subclasses of Ig including IgG, IgG1, IgG2, IgA, IgA1, IgA2 and others. Limited work has been done to investigate the humoral immune response to NiV infection in humans, mainly due to inadequate clinical samples that span the entire disease course of infection and the lack of samples from fatal cases for comparison. Nevertheless, the adaptive immune responses to NiV infection during acute and convalescent phases of two survivors in the 2018 NiV outbreak in India were described [103]. Serum from one of the NiV survivors was found to contain measurable NiV-specific IgG and IgM antibodies within a week after exposure, and the clearance of NiV from blood indicated the elicitation of virus-specific IgG in response to the viral infection. Both NiV survivors had elevated counts in B lymphocytes, which correlated with the generation of NiV-specific IgM and IgG antibodies. This demonstrated that adaptive immune response afforded protection against NiV infection in both acute and convalescent phases of infection; however, the specific antigens that could stimulate the generation of antigen-specific antibodies remain to be identified.

The development of antibodies is also associated with protection against NiV in animals, as well as for viral clearance and recovery. NiV infects a wide range of animals, including fruit bats, pigs, horses, cats and dogs. Studies on the manifestations of NiV in these animals can provide information on the viral pathogenic processes and cellular antiviral responses. For instance, as a reservoir host of NiV, bats have evolved to counteract the immune modulatory effects of viral proteins. It was found that bats have a relatively large repertoire of naive immunoglobulin with high specificity. With larger naive antibody repertoires, bats could control virus replication by the direct clonal expansion of B lymphocytes without the need for immediate affinity maturation to generate high antibodies titers. In addition, bats develop weaker immune responses and delimited production of antibodies as a result of deprived hypermutation and affinity maturation stages of B cells [104]. These immune response features could contribute to the delay in viral clearance and persistence of NiV in bats for a considerably long period. These features could also explain the seroconversion pattern that was observed in the experimentally infected bats with henipaviruses, whereby only 50% of the bats seroconverted, and there were relatively low titers of neutralizing antibody detected [21]. As NiV spillover from bats is possible, leading to potential human NiV outbreaks, it is important to further investigate and identify how immune responses in bats control the viral infection, as well as to compare the immune responses in bats and humans. The findings could provide valuable data on the mechanism of protective immunity against NiV.

With the limitation of human samples to better understand humoral immune response in NiV infection, animal models were established and utilized. In experimentally infected swine, neutralizing antibodies were detectable as early as a week post-infection, followed by high titers of neutralizing antibodies, which developed at two weeks post-infection [105]. Despite the presence of neutralizing antibodies, viral RNA was still detectable in the serum of the swine up to a month post-infection, indicating a slow clearance of NiV in the infected swine. African green monkeys (AGMs) were used as another animal model for NiV infection, where an exhaustion of B cells at 12 days post-infection was noted [106]. The decrease of B cell population over the course of the acute NiV disease correlated with the rapid disease progression. In the study, five out of six AGMs succumbed to NiV infection. The only AGM that survived the infection developed an IgM response and low level of neutralizing antibody after 12 days post-infection. The IgM titer then peaked at 14 days post-infection, and at the same time, an NiV-specific IgG response became apparent. The increase in NiV-specific antibodies in this survived AGM correlated with an increase of the B cell population. A similar delay in the clearance of virus was observed with measles virus (MeV), a virus from the same Paramyxoviridae family [107]. This suggests that transient host immunosuppression and slow clearance of viral RNA could be a feature in NiV infection. Nevertheless, the observation of NiV-infected animals developing antibodies (IgM/G), as well as increase in the population of B lymphocytes, demonstrates that humoral immune responses are activated upon NiV infection as part of the adaptive immunity.

### 7.2. Cellular Immunity—T Lymphocytes

The adaptive cellular immune response is key for the control and clearance of an infection [108]. Cellular immunity is mainly driven by mature T cells, macrophages and the release of cytokines. During an infection, naive T cells are activated into effector T cells, helper T (Th) cells or CD4+ cells and cytotoxic T cells or CD8+ T cells upon exposure to antigenic peptides loaded on major histocompatibility complex (MHC) Class II and I, respectively (Figure 3). Both T cells will respond to MHC molecules that are attached on antigen-presenting cells (APC) and additionally all nucleated cells for CD8+ T cells via the T cell receptor (TCR). The main difference between CD4+ and CD8+ T cells boils down to their primary roles and functions in the immune regulation of the host. The CD4+ T cells trigger immune response via the activation and induction of other immune cells, such as B cells and CD8+ T cells, through the release of cytokines. On the other hand, CD8+ T cells induce cell death by apoptosis or cell lysis via degranulation.

In NiV infection, there are limited reports available on the protective adaptive immune responses upon infection in humans, again due to limitations of human samples. Information, if available, has primarily reported the T cell responses in animal models. Nevertheless, a recent study described the T cell populations during NiV acute and convalescent phases of infection in two human survivors [103]. The absolute number of T cells was noted to remain normal in the blood, but with a significant increase of activated CD8 T cells expressing granzyme B, Ki67 and PD-1. The findings suggested the importance of elevated lymphocyte population, especially cytotoxic effector cells for the elimination of NiV-infected cells.

Similar to the B cell studies, animal models were used for T cell studies as an alternative to human studies to circumvent the limitation of human samples that are available for in-depth studies of NiV pathogenesis. For instance, a similar observation to the human study was observed in two animal experimental studies using AGM and swine [103,106,109,110]. Peripheral immune analysis of the NiV-infected AGMs showed an increase of CD4+ and CD8+ effector memory cells, which correlated with an increase in cytokines and chemokines such as Ki67 [106,109]. Meanwhile, the upregulation of CD25 on Th (CD4+ CD8+) memory cells and CD4− CD8+ cytotoxic T cells was detected in a group of NiV-infected swine [110]. Further analysis showed low levels of viral RNA, and no infectious virus was present in the tissues of the infected swine. Together, these findings highlight the importance of cellular immunity for viral clearance and in surviving NiV infection.

Mice models were also used to examine the adaptive immune response to NiV infection. Balb/c mice were immunized with recombinant avirulent NDV expressing the NiV F protein (rLa-NiVF) [111]. A significant NiV F protein-specific CD8+ T cell response was observed after the first dose, and the response was further boosted after the second dose, unequivocally suggesting the potentiality of this candidate vaccine against NiV infection. Additionally, the first CD8 T cell epitope of the viral F protein (F280) was identified in this Balb/c mouse model, which warrants further T cell studies in animals. In another study, a candidate vaccine using Modified Vaccinia virus Ankara (MVA) to express soluble NiV G triggered a strong NiV G epitope-specific CD4+ and CD8+ T cell response in mice [112]. Several potential epitopes were identified, including the H2-IAb-restricted epitope and H2-b-restricted epitope, which stimulated CD4 and CD8 T cells specific for NiV G.

NiV is a member of the Paramyxoviridae family, which consists of viruses such as MeV, mumps (MuV), RSV and human parainfluenza virus (HPIV), but NiV is most closely related to HeV [5]. Due to the shared characteristics and behavior of paramyxoviruses at the molecular level, the cross reactivity of T cells between the viruses within the family could occur. A recent study supported this hypothesis, as it was found that previous infection with a common human paramyxovirus induced cellular cross-reactivity [113]. Two fusion protein epitope specific-T cell clones (TCCs)—CD8^Xreact^1 and CD8^Xreact^2—were isolated from healthy donors who were previously exposed to MeV and HPIV. The epitope F^129−37^ was found to be highly conserved among members of the Paramyxoviridae family. Concomitantly, when the TCCs were co-cultured with NiV-infected cells, they completely eradicated the infected cells. This confirmed the functionality of the TCCs and suggested that the broadly reactive T cells against other antigenically related members of the Paramyxoviridae family have potential to recognize and offer protection against NiV.

## 8. Immunomodulatory Impact Targets—Cytokines

Regardless of the different but complementing roles of the innate and adaptive immune systems, both the immune subsystems share at least one group of proteins that play a key role in fighting off infections: cytokines. Cytokines are signaling molecules produced by a plethora of types of cells and primarily belong to the innate and adaptive immune systems in response to the presence of foreign substances including viral proteins or antigens. These cytokine molecules comprise smaller groups of proteins such as chemokines, lymphokines, interleukins, interferon and tumor necrosis factors. These proteins mediate various processes such as chemotaxis, tissue repair and cellular proliferation, but most importantly, they help to regulate the inflammatory responses [114].

As previously mentioned, different inflammatory cytokines are induced at different stages and locations in the host during infection, whereby their release could contribute to the worsening of clinical symptoms, such as an increase of vascular permeability, and consequently promote viral spread [115]. An example of an inflammatory cytokine is the chemokine CXCL10, also known as interferon gamma-inducible peptide, IP-10. While CXCL10 helps to promote leukocyte trafficking to the site of infection to generate an inflammatory immune response, it has been shown to cause neurotoxicity [116]. Overexpression of CXCL10 in the brain was shown to lead to neuronal apoptosis via the indirect activation of caspase-3 and calcium dysregulation, whereby the latter subsequently caused cell death [117,118]. In NiV infection, it was suggested that the expression of CXCL10 mRNA followed NiV replication closely. CXCL10 expression was observed in several organs, especially the brain of NiV-infected golden hamster; thus, CXCL10 was suggested to play a role in the development of NiV-associated encephalitis. This is consistent with the findings in the brain epithelial cells of patients who died of NiV-associated encephalitis in the Malaysia NiV outbreak. The brain epithelial cells were noted to be intensely stained with CXCL10. This finding supported the idea that the presence of chemokine in NiV infection is correlated to brain injury and fatality. High expression of CXCL10 was also detected in the lung and spleen tissue of HeV-infected bats. However, NiV/HeV infections are not lethal in pteropid bats. Thus, the high expression of the chemokine suggested a protective role instead, ensuring survival through the infection [119]. Regardless, CLCX10 is a potential marker for lethal NiV-associated encephalitis and could be an excellent target for NiV therapeutics [116].

In addition to CXCL10, the expression of pro-inflammatory cytokines such as IL-6, IL-8, G-CSF, GM-CSF and MCP-1 was also observed in response to NiV infection, particularly in the lungs. In summary, IL-6 induces dendritic cell maturation, which is key for T cell maturation [120]; IL-8 helps to induce the chemotaxis of granulocytes toward sites of infection [121], and MCP-1 helps in regulating blood–brain barrier permeability [122]. The expression of these cytokines, including CXCL10, as shown in NiV-infected primary endothelial cells in vitro, could promote the induction of functional monocytes and T cell movement to the site of infection [75,123,124]. As a result of this increased inflammatory activity, pathological vasculitis was observed, similar to that exhibited in NiV-infected humans.

Although inflammatory cytokines could bring harm to the host, they can also bring benefits when their expression is regulated and balanced. As an example, in addition to initiating dendritic cell (DC) maturation, IL-6 also amplifies the antigen presentation function of DCs, thus improving the efficiency of the initiation of antigen-specific immune responses [122,125]. This function is further improved with Galectin-1, an innate immune effector protein, which is increased in expression at inflammation sites. As DCs would be exposed to high levels of Galectin-1, findings have shown that Galectin-1 upregulates the expression of IL-6 in monocyte-derived DCs [125], which implies the role of Galectin-1 and DCs in mediating an innate inflammatory response for protection against NiV infection.

## 9. Therapeutics and Vaccines—Host and Immune Responses

In a conventional context, antibody response has been associated with immunological measures of vaccine efficacy. While neutralizing antibodies elicited by vaccines are thought to be highly specific and effective, purified antibodies from convalescent serum could be equally efficacious when used in passive immunization, and have been explored as antiviral strategies against henipaviruses. A cross-reactive human monoclonal antibody (m102.4) was developed and has been shown to have a neutralizing effect for both NiV and HeV, in vitro and in vivo [126]. NiV and HeV infection starts with the viral G protein attaching to the host cell receptor (ephrin-B2 or -B3), which then triggers conformational changes to enable the viral F protein to interact and fuse with the host cell membrane [31,48]. Both the NiV F and G proteins are recognized as key antigenic sites for NiV and HeV and are important vaccine candidates. The neutralizing m102.4 binds to NiV G and neutralizes it, thereby blocking the binding of G to the host cell receptor [127]. The effectiveness of m102.4 has been shown in vivo in ferrets and AGMs, whereby the animals were protected against lethal NiV disease after m102.4 was administered into the animals post-exposure to NiV [128,129]. In addition, m102.4 was evaluated in a phase 1 human study, and it was noted that a single and repeated dose of m102.4 was safe and well tolerated [130]. Another recent therapeutic development for NiV disease is the cross-reactive humanized mAb h5B3.1 targeting the NiV F protein. The antibody showed promising protection from NiV and HeV disease in ferrets [131]. Structural analysis of NiV F in complex with the mAb h5B3.1 revealed that the antibody was able to block membrane fusion activity by locking the F protein in a pre-fusion conformation [132]. However, m102.4 is the only human monoclonal antibody that has been evaluated for NiV and HeV protection studies in the AGM model, as well as having undergone phase 1 human study. Collectively, these studies provided the proof-of-concept that monoclonal antibody immunotherapy against NiV infection by targeting the viral glycoproteins could provide protection. This further highlights the importance of humoral response to NiV glycoproteins as a mechanism for protection.

The use of a safe and efficacious vaccine against viral pathogens is one of the main medical countermeasures against viral infection in humans. The key factor that contributes to a vaccine-induced protective mechanism is the production of neutralizing antibodies. For NiV, the viral F and G proteins are the main antigen binding sites for neutralizing antibodies. Therefore, a successful vaccine against NiV will be one that can elicit neutralizing antibodies specific against these viral proteins. Several vaccine strategies have been developed and tested in animal models [133,134,135]. Due to the high mortality of NiV, a safe, live-attenuated vaccine with no potential of reversion is considered a difficult approach. Therefore, most of the NiV vaccine candidates in development are focused on subunit vaccine and live-vectored vaccine approaches. The most extensively studied approach is the recombinant subunit vaccine incorporating the soluble G protein of NiV (NiVsG) or HeV (HeVsG). Both NiVsG and HeVsG have shown promising results in preventing disease following exposure to both NiV and HeV in animal experimental studies, respectively, suggesting that an effective subunit vaccine strategy appears achievable [134,135]. It is also noteworthy that HeVsG showed effective cross-protection against NiV infection in ferrets [135], as the G protein of NiV and HeV shares 83% of amino acid identity [136]. In 2012, HeVsG was developed as the first commercialized horse vaccine against HeV, which was named Equivac and is marketed by Zoetis, Inc., under the Australian Pesticides and Veterinary Medicine Authority (APVMA) [137]. For humans, the HeVsG subunit vaccine is currently in clinical development as an emergency vaccine for NiV outbreaks [138]. In addition, viral vector-based recombinant vaccines carrying NiV F or G protein on their surfaces have been developed. Various viral vectors including rabies virus (RABV), canarypox virus (CNPV), adeno-associated virus (AAV), MeV, NDV, vesicular stomatitis virus (VSV) and Venezuelan equine encephalitis virus (VEEV) have been explored for NiV vaccine development [139,140,141,142,143]. Among the vectored vaccines, a recombinant MeV that expresses the NiV G protein (rMV-NiV-G) has shown promising protection against the disease in AGM, suggesting that the rMV-NiV-G is a promising vaccine candidate for use in human [141]. Besides, an LNP-encapsulated mRNA vaccine encoding the soluble HeV G protein (sHeVG mRNA LNP) was recently developed [144]. In the Syrian hamster model study, sHeVG mRNA LNP showed promising cross-protective results against NiV, with 70% of the animals surviving a lethal NiV challenge [144]. Findings from these candidate vaccine studies have provided a foundation insight for developing vaccines against NiV infections, highlighting the importance of the viral glycoproteins that stimulate neutralizing antibodies as vaccine candidates to protect against the disease, as well as to potentially cross-protect against closely related viruses.

Several antiviral drugs were investigated for NiV treatment. However, only a few of them have been evaluated in animal model studies, such as ribavirin, remdesivir and favipiravir. The first antiviral drug that was used against NiV was ribavirin (Copegus, and others). During the outbreak in Malaysia in 1998–1999, for a total of 140 NiV-infected patients who were treated with ribavirin, a 36% reduction of the mortality rate was found [145]. However, there was no significant reduction of mortality when evaluated in vitro when ribavirin was administered in combination with chloroquine in hamsters [146]. Ribavirin was utilized during the 2018 NiV outbreak in Kerala: six patients received an oral ribavirin, and only two survived [147]. Due to the lack of proven therapy against NiV, more recently, another adenosine nucleoside antiviral drug, remdesivir (Veklury), was tested in AGMs. Mild respiratory symptoms developed in two of four animals treated with remdesivir, while severe respiratory symptoms developed in all the untreated animals, suggesting that remdesivir is a promising antiviral drug against NiV [148]. Recently, remdisivir was approved for use to treat SARS-CoV-2 infection by The United States Food and Drug Administration and has been authorized for emergency use in several countries such as Bangladesh, Singapore, Taiwan, India, Japan and Austrailia [149]. Another antiviral, acyclovir (Zovirax), was used together with ceftriaxone to treat nine abattoir workers during the NiV outbreak in 1999 in Singapore, and eight of them survived [150]. There are no data from in vitro studies of acyclovir against NiV. Favipiravir, which is sold under the brand name Avigan, and others have been shown to inhibit NiV replication in vitro [151]. In addition, favipiravir exhibited the highest antiviral activity against NiV infection in a hamster model study [152]. Rintatolimid (Ampligen), an immuno-modulator, was found to be effective in inhibiting NiV replication and to protect against viral challenge by inducing IFN-α and IFN-β in hamsters [93]. However, more solid evidence on the efficacy of antiviral drugs that could effectively be used for the treatment of NiV infection in humans is required as there are currently limited in vitro and in vivo studies that have been conducted.

## 10. Conclusions

NiV is a zoonotic virus that is associated with high morbidity and mortality in humans and animals. NiV was demonstrated to effectively interfere with both innate and adaptive immune responses and to have mechanisms to suppress the host antiviral response. This review summarizes the current understanding of NiV pathogenesis, innate and adaptive immune responses induced upon NiV infection, as well as the host response to therapeutics and candidate vaccines for NiV. Due to the limitations to what testing can ethically be conducted in human subjects, relevant animal models that can mimic human NiV disease are critical to perform mechanistic studies of clinical observations and for a better understanding of virus pathogenesis. The fact that data from human cases are sparse highlights the importance of findings from studies performed in animal models such as hamsters, AGMs, swine, ferrets and mice that are able to closely recapitulate the clinical signs of NiV disease in human. Several animal experimental models for NiV have successfully been developed and used to evaluate immune responses following NiV exposure. Although the in vitro and in vivo studies demonstrated some commonalities in the response to NiV infection and suggested a potential immune response that correlates with the survival of NiV infection, complete immunopathogenesis and immunological pathways associated with NiV infection in human are largely unknown. There is still no clear evidence and experimental models that can precisely demonstrate human cellular and systemic response in the event of NiV infection and re-exposure. Further studies on the NiV-specific immune responses in humans are still required to provide a robust framework to understand the mechanism of protection against NiV in humans.

## Figures and Tables

**Figure 1 microorganisms-10-01162-f001:**
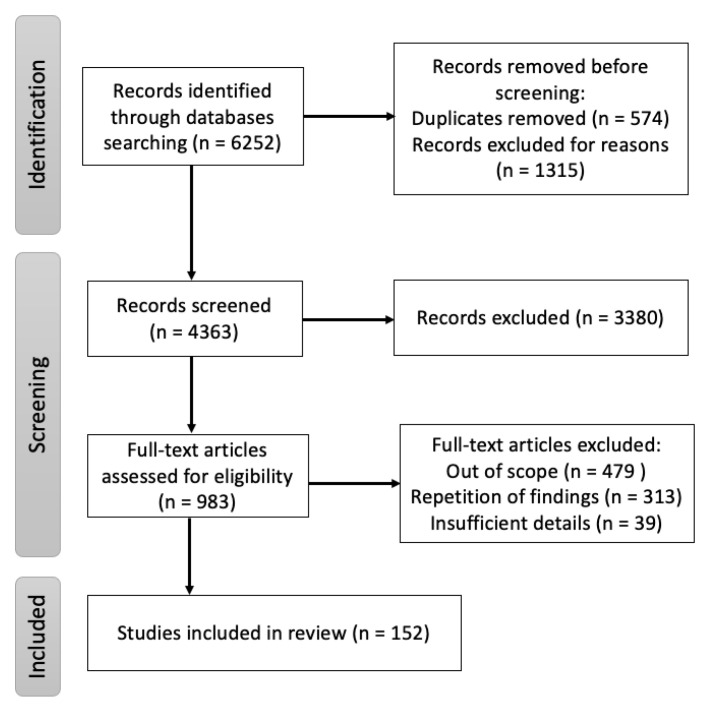
Flow diagram of the review of immune responses, pathogenesis, transmission of the disease in humans and animal models, and medical countermeasures associated with NiV.

**Figure 2 microorganisms-10-01162-f002:**
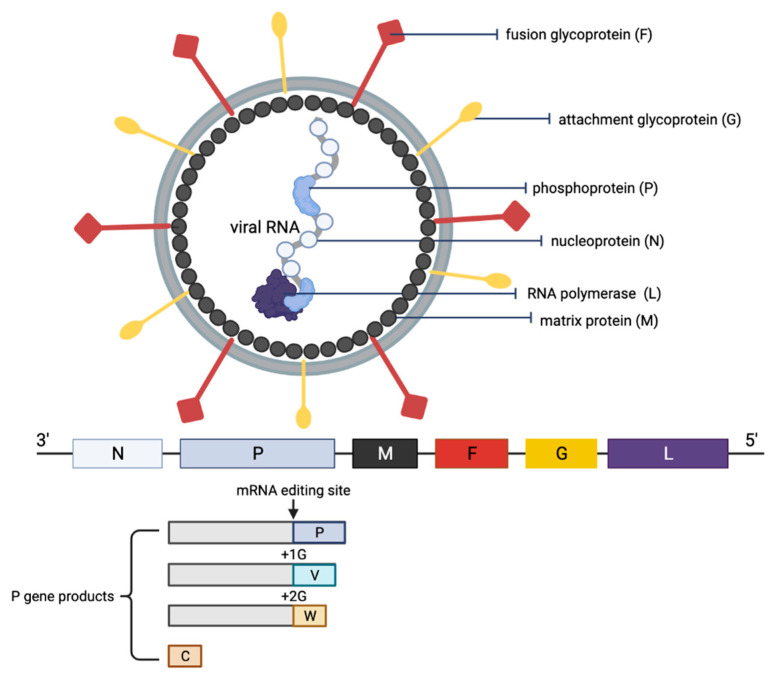
Schematic representation of the structure of an NiV particle and the viral genome organization. The NiV N, P and L proteins interact with the viral RNA to form the ribonucleoprotein complex, which is surrounded by a lipid bilayer envelope containing the NiV glycoproteins F and G. The NiV M protein is associated with the inner side of the envelope. The viral proteins and arrangement of genes in the viral genome from 3′–5′ are color-coded, respectively, for identification. The NiV P gene products (V, W and C proteins) as a result of mRNA editing are illustrated. The V protein contains a single G insertion, and translation shifts it to +1 reading frame. The W protein contains two G insertions, shifting the translation to the +2 reading frame. The C protein is translated from an internal open reading frame of the P gene.

**Figure 3 microorganisms-10-01162-f003:**
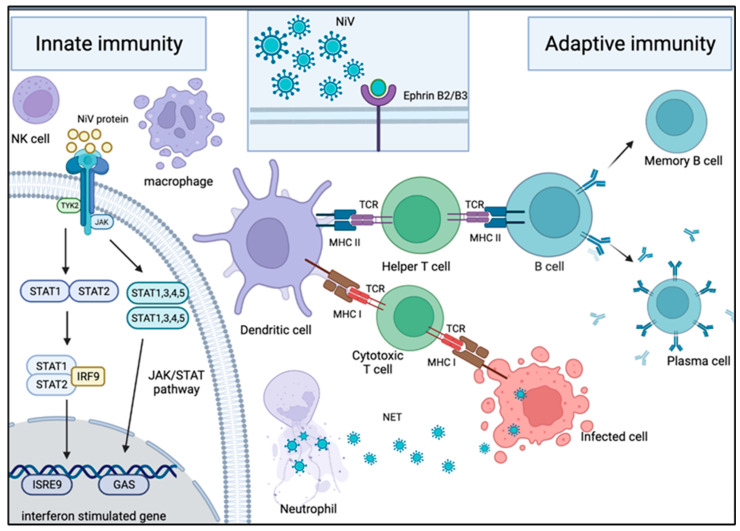
Mechanism of host immune response in NiV infection. *Innate immune response upon NiV infection:* Neutrophils are one of the first immune cells to be recruited to the site of infection. These cells use NET and antiviral molecules to contain the virus in web-like traps. To counteract this, some NiV proteins are capable of inhibiting the IFN response by interacting with the JAK/STAT pathway. *Adaptive immune response upon NiV infection:* Following NiV entry via ephrin-B2 or -B3 receptors on the host cellular membrane, it is engulfed and broken down into viral peptides by APC such as macrophages and dendritic cells. The presentation of the viral peptides on the MHC molecules activates the T cells through their TCR. The activation of the helper T cells subsequently drives B cells to activate, proliferate and develop a mature antibody response. As a result, plasma cells and memory B cells are formed, producing NiV-specific antibodies for protection against infection. On the other hand, the activation of cytotoxic T cells allows them to target and kill the NiV-infected cells.

## Data Availability

Not applicable.

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
