# Peer review of "The Immunobiology of Nipah Virus"

_microorganisms, 2022, doi:10.3390/microorganisms10061162_

Round 1

Reviewer 1 Report

YJ Mei Liew and colleagues present draft of the review paper entitled "The immunobiology of Nipah virus" for consideration for publication in the journal of Microorganisms. In view of recent COVID pandemic the presented topic sounds interesting as well as definitely more attraction of the readers might gain. Despite that the Nipah virus is not so abundant worldwide, it has a high potential to become another trigger for another pandemic breakout in the future. 

After carefully conducted review of this paper I must admit that in its current form it cannot be published, albeit, I strongly encourage authors for extensive revision and resubmission. Please let me explain my major concerns:

Firstly, the Authors aimed to describe recent progress in the state of the knowledge in terms of a broad term - immunobiology. Sadly, I found that most of the cited papers have been published before the year of 2012, so they are definitely outdated. Taking into account recent advances in the field of immunology in general, as well as Nipah virus advances the literature background is not acceptable. Recently (5 years till now) 477 scientific papers indexed in PubMed and focusing on Nipah virus have been published - only few of them cited. Please repeat and enrich all the references. 

Moreover, because of usage of very old references many information and facts are missing - please refer to:

  • nipah virus and tnf-alpha/il-1b interplay
  • vsv-vectored vaccinies discovieries
  • Architecture and antigenicity of the Nipah virus attachment glycoprotein. 
  • Nipah Virus V Protein and Altering MDA5 Helicase
  • Role of caspases, IL-8, CXCL X, G-CSF and other cytokines must be considered as a key point of Nipah virus immunomodulatory impact targets
  • The impact of Nipah virus on kidneys, spleen, liver should be discussed in terms of molecular/metabolic pathways
  • The methods of virus cleavage/engulfing/removing should be discussed
  • novel approaches for virus prevention must be discussed

Moving further, there is no information given regarding the following PRISMA guidelines and description of the searching strategy, which is so crucial for review papers. The Author should include at least: Data sources and searches, Study eligibility criteria, Study selection process, Data extraction, and study quality assessment (assessing the risk of bias (ROB) for each included study), Data synthesis. MeSH terms and keywords are necessary to be included. For each step, it is necessary to explain for the reader with pictures or tables. It is necessary to explain what was drawn at each step to lead to the result. Moreover, a figure showing the PRISMA-based workflow should be drawn. After that, a discussion is valuable.

The figure showing the structure of the virus with its protein composition should be included

Additional figure(s) showing the main molecular pathways involving the virus should be added for better understanding the phenomena

The broad insight into neutrophils/NETs interplay between virus and immune response must be discussed.

Please also include recent clinical trials conducted to prevent Nipah virus infection.

Stages of assembly and replication of the virus in host should be discussed in more in-depth manner

The potential usage of classic antiviral agents should be discussed as well.

Reviewer 2 Report

The submitted manuscript entitled “The immunobiology of Nipah virus” provides a good overview about what is currently known regarding the host immune response to henipavirus infection.  Overall this relatively brief review (appropriately brief given the paucity of data on this topic) summarizes what is known quite well.  This reviewer does not have and significant comments except to suggest that the authors carefully proof the manuscript for grammar.

Round 2

Reviewer 1 Report

The Authors made lots of improvements and substantially increased the quality of the work. Almost all of my concerns have been addressed correctly and adequately fixed throughout the manuscript. 

Only two majors remain - the most important is to put a graph (please see figure 2 in this paper as the reference: https://www.mdpi.com/1648-9144/58/4/472 ) showing PRISMA flow. It is important since all the published review papers must follow strictly PRISMA guidelines.

Secondly, accordingly to COPE guidelines, review and editing, are not considered the substantial input to the paper, thus, most of the Authors' contributions should be mentioned in the Acknowledgment section, not listed as the co-Authors. Please reconsider. The review paper should not list more than 5 authors meanwhile provided paper lists 10 of them. It raises ethical concerns. 

Author Response

Point 1: Only two majors remain - the most important is to put a graph (please see figure 2 in this paper as the reference: https://www.mdpi.com/1648-9144/58/4/472 ) showing PRISMA flow. It is important since all the published review papers must follow strictly PRISMA guidelines.

Response 1: Agreed, the Figure has been added under the Methods subheading, line 119-122.

Point 2: Secondly, accordingly to COPE guidelines, review and editing, are not considered the substantial input to the paper, thus, most of the Authors' contributions should be mentioned in the Acknowledgment section, not listed as the co-Authors. Please reconsider. The review paper should not list more than 5 authors meanwhile provided paper lists 10 of them. It raises ethical concerns.

Response 2: The Authors Contributions has been revised. All authors contributed to the writing and editing, particularly in their areas of expertise; C.T.T. and J.P.S. on clinical disease, pathogenesis and therapeutics; and R.G.R., N.G.C. and W.F.W. on the immunology, therapeutics and vaccines.